# Spatiotemporal motion prediction in free-breathing liver scans via a recurrent multi-scale encoder decoder

**Author(s) names withheld**                                    EMAIL(S) WITHHELD

## Abstract

In this work we propose a multi-scale recurrent encoder-decoder architecture to predict the breathing induced organ deformation in future frames. The model was trained end-to-end from input images to predict a sequence of motion labels. Targets were created by quantizing the motion fields obtained from deformable image registration. We propose a multi-scale feature extraction scheme in the spatial encoder which processes the input at different resolutions. We report results using MRI free-breathing acquisitions from 12 volunteers. Experiments were aimed at investigating the proposed multi-scale design and the effect of increasing the number of predicted frames on the overall accuracy of the model. The proposed model was able to predict vessel positions in the next temporal image with a mean accuracy of $2.03 \pm 2.89$ mm showing increased performance in comparison with state-of-the-art approaches.

**Keywords:** motion prediction, liver, MRI, free-breathing, LSTM

## 1. Introduction

According to the last American Cancer Society's report (Ame, 2020), about 42,810 new cases of primary liver cancer will be diagnosed this year in the US. Radiation therapy is the first line of treatment for the majority of these cases. Its goal is to focus the radiation beams in the target and to avoid surrounding anatomy. However, respiratory motion is one of the major issues with large dosimetric impact (Mechalakos et al., 2004). Image-guided radiation treatments can greatly benefit from future frame prediction models since the beam can be re-positioned compensating for motion. Toward this end, several solutions have been proposed. Generally, they rely on statistical (Samei et al., 2012; Preiswerk, 2013) or biomechanical modeling (Brock et al., 2002), the former being more common in the literature. In this work, we proposed a recurrent multi-scale encoder-decoder framework to perform in-plane spatio-temporal motion prediction from sequential images.

## 2. Method

The proposed model aims at learning a representation that predicts the sequence of encoded motion $\langle \boldsymbol{Z_n}, \boldsymbol{Z_{n+1}}, \ldots, \boldsymbol{Z_{n+T}} \rangle$ over $T$ future time steps given an input image sequence $\langle \boldsymbol{I_1}, \boldsymbol{I_2}, \ldots, \boldsymbol{I_n} \rangle$ of length $n$. Figure 1 (A) shows the proposed pipeline. First, consecutive pair of images are non-rigidly registered in order to measure the motion between them. Secondly, the resulting two-dimensional motion fields are encoded using an auxiliary representation space $\boldsymbol{Z_i} = \mathscr{F}(\boldsymbol{Y_i})$ where $\boldsymbol{Z_i} \in \mathbb{R}^{H \times W \times Q}$. $\mathscr{F}$ is a mapping function to encode the

displacement fields into motion labels. To that end, the ranges of values for each vectorial component, i.e. axes $x$ and $y$, are quantized into $b$ bins according to the data distribution. A codebook $\mathbf{C} \in \mathbb{R}^Q$ is built by assigning a class to each possible combination between the bins of each axis. Then, the architecture presented in Figure 1 (B) is trained from input and target sequences. It contains a multi-scale (MS) spatial encoder that extracts feature representations at multiple scales through the network as showed in Figure 1 (C). The MS block processes the input tensor at different levels: fully resolution, medium resolution and low resolution in order to fully exploit the image features. The motion learning architecture also contains recurrent units and a fully convolutional spatial decoder. The spatio-temporal features extracted by the MS encoder are extrapolated in time by the convolutional Long Short-Term Memory (LSTM) units and further processed by the spatial decoder to recover the desired dimensions in the form of motion labels.

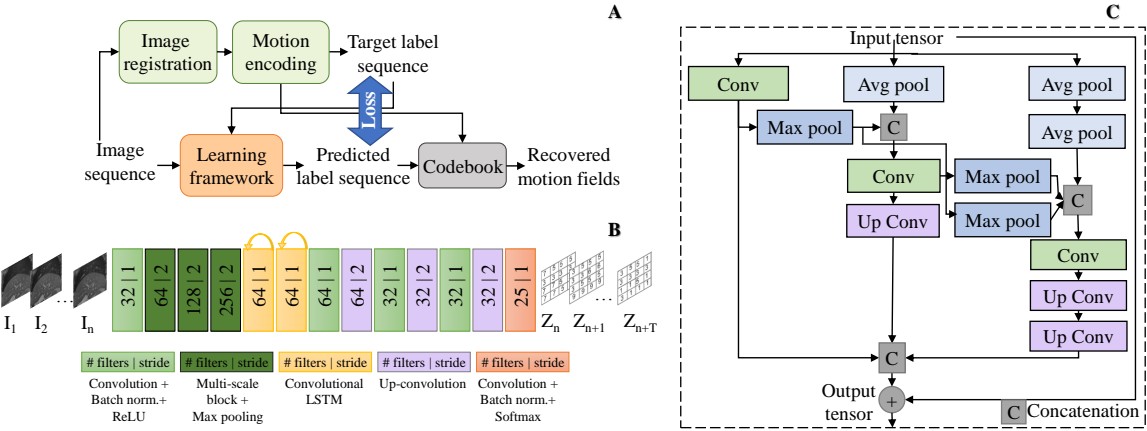

Figure 1: A) Proposed pipeline B) Motion prediction architecture C) Multi-scale block.

We introduced a weighted cross entropy loss function (Zhang et al., 2016) to promote class rebalancing since the distribution is strongly biased toward classes representing the superior-inferior motion:

$$L_{CE} = -\sum_{h,w} v\left(\mathbf{Z}_{h,w}\right) \sum_{q} \mathbf{Z}_{h,w,q} \, log\left(\hat{\mathbf{Z}}_{h,w,q}\right), \qquad (1)$$

where $v\left(\mathbf{Z}_{h,w}\right) = \mathbf{w}_{q^*} \propto \left((1-\lambda)\,\tilde{\mathbf{p}}_{q^*} + \frac{\lambda}{Q}\right)^{-1}$, with $\tilde{\mathbf{p}}_{q^*}$ the empirical distribution of motion class $q^*$ : $\sum_q \tilde{\mathbf{p}}_q \, \mathbf{w}_q = 1$, $q^* = \underset{q}{argmax} \, \mathbf{Z}_{h,w,q}$ and $\lambda$ is the smoothing weight.

We split each volunteer dataset in 60/20/20 for training, validation and testing, respectively. Adam optimizer with an initial learning rate of $10^{-3}$ was used. This learning rate was reduced by 2 after 10 epochs without improvements in the validation set accuracy.

## 3. Results and discussion

Experiments were conducted using 50 MRI dynamics covering 15 positions on the right liver lobe from 12 volunteers. Pixel spacing, slice thickness and temporal resolution are

equal to $1.7 \times 1.7$ mm$^2$, 3 mm and 320 ms, respectively. We compared the proposed network with statistical modeling (Li et al., 2011) and with a similar architecture which uses the traditional encoding scheme (Conv-Pool stacking) (Luo et al., 2017). Table 1 presents the mean landmark location errors in the predicted images with respect to ground truth positions. The proposed model significantly outperforms the compared methods for the in-plane motion prediction task. Figure 2 presents a comparison based on the Normalized Cross Correlation (NCC) metric. Higher values, which were obtained with the proposed method, indicate a greater spatial correlation. Figure 3 illustrates the vessel trajectory through the target and predicted temporal images. Our multi-scale encoder-decoder model showed the closest alignment with the target trajectory. Finally, Figure 4 shows an example of the output sequence obtained by deforming the last input image with the predicted deformations. Although the last three extrapolated images present some degree of misalignment with the targets, we can still identify the vessels and track the anatomy.

Table 1: Vessel tracking error position (in mm) for each predicted time in the MRI dataset. Values are mean $\pm$ standard deviation.

| Model | t=1 (320 ms) | t=2 (640 ms) | t=3 (960 ms) | t=4 (1280 ms) | t=5 (1600 ms) |
|---|---|---|---|---|---|
| PCA | $4.38 \pm 5.12$ | $4.64 \pm 4.76$ | $5.03 \pm 5.19$ | $5.39 \pm 5.42$ | $5.86 \pm 5.73$ |
| Enc-Dec | $2.68 \pm 3.22$ | $3.36 \pm 3.37$ | $3.90 \pm 3.58$ | $4.39 \pm 3.61$ | $4.38 \pm 3.42$ |
| Proposed | $2.03 \pm 2.89$ | $2.93 \pm 3.26$ | $3.51 \pm 3.77$ | $3.79 \pm 3.92$ | $3.86 \pm 3.89$ |

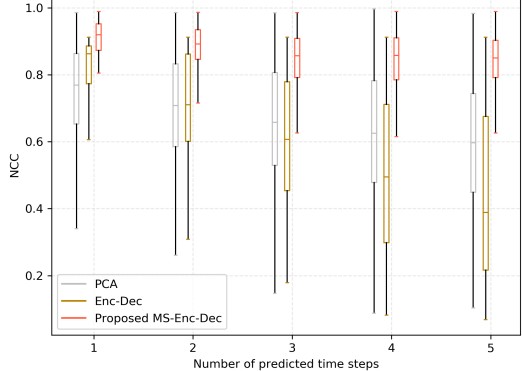
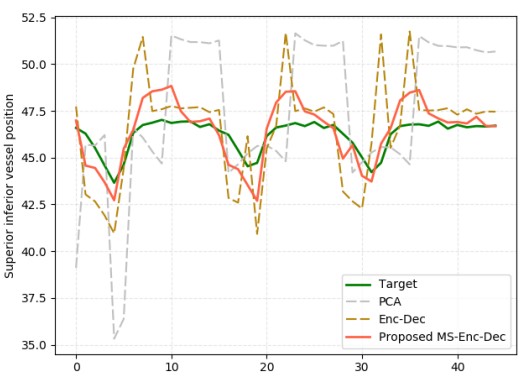

Figure 2: NCC between target and predicted images.

Figure 3: Vessel trajectory predicted with different approaches.

# References

*Facts and Figures 2020.* American Cancer Society, Atlanta, Ga., 2020.

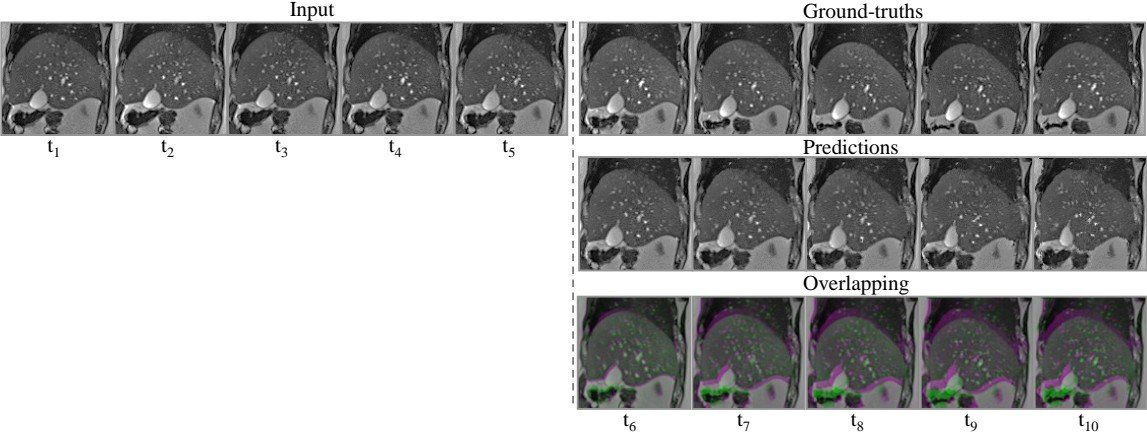

Figure 4: Extrapolation up to five time steps from an input sequence in the test set. Top row: Input and target sequences, middle row: predicted images by the proposed model, bottom row: overlapping between target and predicted images. Green and magenta pixels belong to target and predicted images, respectively.

KK Brock, SJ Hollister, LA Dawson, and JM Balter. Creating a four-dimensional model of the liver using finite element analysis. *Medical physics*, 29(7):1403–1405, 2002.

Ruijiang Li, John H Lewis, Xun Jia, Tianyu Zhao, Weifeng Liu, Sara Wuenschel, James Lamb, Deshan Yang, Daniel A Low, and Steve B Jiang. On a pca-based lung motion model. *Physics in Medicine & Biology*, 56(18):6009, 2011.

Zelun Luo, Boya Peng, De-An Huang, Alexandre Alahi, and Li Fei-Fei. Unsupervised learning of long-term motion dynamics for videos. In *Proceedings of the IEEE Conference on Computer Vision and Pattern Recognition*, pages 2203–2212, 2017.

James Mechalakos, Ellen Yorke, Gikas S Mageras, Agung Hertanto, Andrew Jackson, Ceferino Obcemea, Kenneth Rosenzweig, and C Clifton Ling. Dosimetric effect of respiratory motion in external beam radiotherapy of the lung. *Radiotherapy and Oncology*, 71(2): 191–200, 2004.

Fran Preiswerk. *Modelling and Reconstructing the Respiratory Motion of the Liver*. PhD thesis, University of Basel, 2013.

Golnoosh Samei, Christine Tanner, and Gabor Székely. Predicting liver motion using exemplar models. In *International MICCAI Workshop on Computational and Clinical Challenges in Abdominal Imaging*, pages 147–157. Springer, 2012.

Richard Zhang, Phillip Isola, and Alexei A Efros. Colorful image colorization. In *European conference on computer vision*, pages 649–666. Springer, 2016.

