# OpenReview forum: "Spatiotemporal motion prediction in free-breathing liver scans via a recurrent multi-scale encoder decoder"
_MIDL.io/2020/Conference — MIDL 2020_

### Official Review · AnonReviewer2 · 2020-03-01
**MRI motion prediction via recurrent MS encoder-decoder - Review**

**Rating:** 3
**Confidence:** 3

**Review:**

As the manuscript title suggests, the authors propose applying an encoder-decoder architecture originally introduced for motion dynamics learning in computer vision context to motion prediction in 2D(+t) liver MR imaging. Following the description in the manuscript, in contrast to the corresponding CVPR 2017 publication, the authors introduce a multiscale block to extract feature from three different spatial scales. The actual temporal prediction is performed by a convolutional LSTM.
The short paper is well structured and an interesting read. Indeed, I would have liked to read more about details about the applied method (eg. the exact motivation and setup of the codebook) – but the page limit recommendation for short papers is in some contradiction to this.

Overall assessment:
The contribution transfers and adapts a CVPR 2017-published methodical paper to the medical imaging domain (here: spatio-temporal MR imaging). Thus: One can argue that the methodical novelty of the contribution is limited; I nevertheless like the paper.

Minor aspects:
- Fig. 2 and Table 1  seem to be in contradiction (at least in parts): While NCC for Enc-Dec is worse than PCA, the corresponding LM tracking errors show a different picture. This needs some explanation.
- The discussion states that the proposed model *significantly* outperforms the other approaches. How was this statistically tested?
- The last sentence of the discussion is irritating: If one can identify the vessels and anatomical positions but at erroneously predicted positions, why should this be helpful?

---

### Official Review · AnonReviewer4 · 2020-03-12
**The authors present interesting ideas to predict deformation in future images. The method is still at the beginning of its development and there are a lot of things to work on. However, I think it might be helpful to present this work at MIDL and discuss further developments.**

**Rating:** 3
**Confidence:** 4

**Review:**

Summary:

The authors propose a multi-scale encoder-decoder architecture to predict breathing induced organ 2D deformation in future frames. They train and test on just 12 MR sequences of unknown origin. In the evaluation, they compare their method with two other methods and show that their method performs best. For me, a few important explanations and experiments are missing in this paper.
Overall, I think the authors present interesting ideas to predict deformation in future images. The method is still at the beginning of its development and there are a lot of things to work on. However, I think it might be helpful to present this work at MIDL and discuss further developments.


Pros:

•    The authors deal with the difficult question of motion prediction in future images. To reduce the search space of possible motions, first, they analyse the motion in the training sequence and quantized them into b bins and thereby convert the regression task into a classification task. I think this is an interesting way to handle this difficult task.
•    The paper is well written and mostly easy to follow.


Open Questions:

•    What is Q?
•    Which image registration method is used to align the images? How good is this method?
•    Are the authors *introducing* or *using* the weighted cross-entropy loss function?
•    Where are the data from? What kind of manual annotation are available? (How many landmarks on which positions?)
•    How good is the mean landmark location error if the identity is used as the deformation field?
•    If I understand correctly, one training per patient is needed. So for a clinical application, you first have to acquire a sequence of a patient to train the network. Afterward, the trained network can be applied during the treatment.  How long does the training take? Are results worse when there is a longer break between training and inference?
•    What is the runtime of the method during inference?


Cons:

•    The authors don’t stick to the 3-pages limit.
•    After reading the paper, I still have open questions that should be answered in the paper.
•    The mentioned related work is quite old (from 2002,2012,2013). I would assume that in the last past seven years, people have worked on this topic as well.
•    If I understand correctly, no regularisation of the deformation field is used. Does the method generate smooth deformation fields without foldings? An analysis of the volume changes and foldings is missing.
•    Are the authors *introducing* or *using* the weighted cross-entropy loss function?
•    In Figure 3, the time axis doesn’t have a unit.

---

### Official Review · AnonReviewer3 · 2020-03-19
**Interesting problem**

**Rating:** 2
**Confidence:** 3

**Review:**

The paper presents encoder-decoder architecture to predict motion for liver MRI.
First, the authors generate displacement field between pair of images using (unknown) registration framework, then encode this displacement field into label, and generate a codebook between the label and a quantized vectorial components of the displacement field.
Secondly, the authors train the network (decoder + LSTM) to predict the motion label, and finally use the codebook to recover motion field.
the method is validated using 50 MRI scans (2D) coming from 15 volunteers, the vessel tracking error is given for the presented method, and two other relevant methods, showing improvement accuracy for the presented method.

Pros:
- (real time?) motion estimation for mri guided therapy is really emerging problem, and so the presented approach is interesting contribution

cons
- the approach consists of several steps, while it is not really clear whether they are needed.
- Would be possible to train encoder-decoder to predict directly motion from the sequence?
- What extra quantization add to overall accuracy?

since this is 2D(?) acquisition, and the problem described is the breathing motion, is there any issue with out-of-plane motion? Could this explain rather large error at the end of sequence?

It is also not clear whether this acquisition is 2d or 3d. Page 1 says that the registration is done between images to produce 2d motion field, then the data description (Page 3) says pixel spacing, and slice thickness. Is 3D MRI and split into 2d slices?

testing/validating
- 50 MRI scans from 12 volunteers. Were the same volunteer scans used for training and testing?

- the authors wrote that the results are significantly better, no test, p-value given

- what is vessel tracking error?
- how many landmarks were used?


More general problem to consider:
What registration between pair of images was used?
There has been a bit of research done on (both MRI and CT) liver motion estimation using discontinuous registration ("A locally adaptive regularization based on anisotropic diffusion for deformable image registration of sliding organs." IEEE transactions on medical imaging 32.11 (2013): 2114-2126. "GIFTed Demons: deformable image registration with local structure-preserving regularization using supervoxels for liver applications." Journal of Medical Imaging 5.2 (2018): 024001.)

---

### Official Review · AnonReviewer1 · 2020-03-20
**Incorrect training / test / validation split.**

**Rating:** 1
**Confidence:** 3

**Review:**

The paper proposes a recurrent multi-scale architecture for motion prediction in free-breathing MRIs.

* The paper says: "We split each volunteer dataset in 60/20/20 for training, validation and testing, respectively." It appears that each of the 12 volunteer's images was split in "60/20/20" along the time-axis and included in each of the training, validation and test sets. I don't think this is the right way to split the data. The training / test / validation sets should contain images from different subjects. With the current data split, I don't think the results can be trusted.

* Also, several things in the description of the method are unclear to me:
    - what is the difference between 'displacement fields', 'motion fields' and 'motion labels'.
    - "To that end, the ranges of values for each vectorial component, i.e. axes x and y, are quantized into b bins according to the data distribution." Which data distribution?

---

### Meta-Review · Area_Chair1 · 2020-04-03
**MetaReview of Paper57 by AreaChair1**

**Rating:** 3

**Metareview:**

Two reviewers expressed substantial interest and value of the approach for motion prediction, one was somewhat interested and the last one provided a very short review at the last minute. I agree that the paper is neither fully comprehensible nor well validated, but given the page limit the authors present an interesting idea that might deserve further discussion. It should be possible to slightly improve the clarity given the very detailed and comprehensive reviews. In particular the question how exactly the quantisation of motion fields was performed and how it relates to discrete displacement registration would be an important one to answer in the final version.

**Paper Type:**

methodological development

---

### Decision · Program_Chairs · 2020-04-11

Accept